# Relationship between Built Environment and COVID-19 Dispersal Based on Age Stratification: A Case Study of Wuhan

**DOI:** 10.3390/ijerph18147563

**Published:** 2021-07-16

**Authors:** Qiang Niu, Wanxian Wu, Jie Shen, Jiaxin Huang, Qiling Zhou

**Affiliations:** 1School of Urban Design, Wuhan University, Wuhan 430072, China; niuqiang@whu.edu.cn (Q.N.); 2020282090084@whu.edu.cn (W.W.); 2School of Urban Construction, Wuhan University of Science and Technology, Wuhan 430072, China; huangjiaxin@wust.edu.cn; 3School of Foreign Languages and Cultures, Chongqing University, Chongqing 400044, China; 20190755@cqu.edu.cn

**Keywords:** COVID-19, built environment, young and middle-aged, elderly, Wuhan

## Abstract

The outbreak of COVID-19 (coronavirus disease 2019) has become the focus of attention in the field of urban geography. Built environment, such as the layout of public spaces like transportation hubs and urban open spaces, is an important factor affecting the spread of the epidemic. However, due to the different behavior patterns of different age groups, the intensity and frequency of their use of various built environment spaces may vary. Based on this, we selected patients that were infected, with a non-manipulated time period, and the classification of human behavior patterns; we then conducted a regression analysis study on the spatial distribution and building environment of these COVID-19 patients. The results showed that the spatial distribution of young and middle-aged patients (18–59 years old) was more homogeneous, while the spatial distribution of elderly patients (60 years old and above) had a strong clustering characteristic. Moreover, the significant built environment factors exhibited in the two populations were extremely different. More diverse urban facilities and public spaces exhibited influential properties for older patients, while middle-aged and young adults were more influenced by commuting facilities. It can be said that the built environment shows different influences and mechanisms on the transmission of respiratory infectious diseases in different populations. Therefore, the results of this paper can inform decision makers who expect to reduce the occurrence of urban respiratory infectious diseases by improving the urban built environment.

## 1. Introduction

The COVID-19 epidemic, discovered and reported at the end of 2019, has been spreading around the world successively in 2020, which has posed a serious threat to global public security. The outbreak was identified as a “global pandemic” on 11 March 2020. The rapid and large-scale spread of COVID-19 epidemic involves viruses, humans, the environment and other factors, which have brought great challenges to the ability of all sectors of society to quickly respond to public safety and health events. Meanwhile, it has become the focus of common attention in public health, sociology, urban geography and many other fields [1]. Public health focuses on the origin of the virus and the mechanism of infection [2,3,4], clinical features of the patient [5,6,7], complications [8] and other issues. Attention is also given to drug approaches to controlling the virus [9,10,11], the development of specific vaccines [12,13] and the allocation of medical resources [14]. Sociology focuses on people’s anxiety, depression and social inequality in the epidemic period [15,16]. Urban geography mainly studies the spatial–temporal distribution and evolution of the virus [17,18,19,20,21,22], the influence of urban spatial environment factors on the spread of the epidemic and the optimization strategy of urban spatial layout in the post-epidemic era [23,24,25] aiming to build an effective and defensible urban spatial system.

According to studies in the field of public health, COVID-19 is mainly transmitted through respiratory tract, and the transmission efficiency of the virus is greatly affected by air circulation and air quality [26], so that face-to-face contact and social interaction often lead to the spread of the disease. In areas such as cities, with high population density and high demand for outdoor activities, where spatial contact between residents and the built environment occurs frequently, the risk of transmission of respiratory infections increases [26,27], which led to the initial outbreak of COVID-19 in large cities.

The relationship between urban spatial pattern, built environment and COVID-19 dissemination is the core topic of urban geography [28]. Among them, the urban built environment is divided into multi-angle and multi-factor research. Many scholars have studied built environment (for example, building density, hospital density, public green space density and other factors) and epidemic transmission. Previous studies have typically investigated the correlation between the built environment and the spatial distribution of patients by constructing regression models [17], GWR (geographically weighted regression) models, or SLM (spatial lag model) [19]. Most of them mainly focus on the spatial distribution of different built environment factors and the epidemic situation, or only study one of the factors in depth. Li proposed and verified that six factors, including urban growth, hospital density, commercial facilities, subway stations, mixed land use and urban population aging degree were positively correlated with the severity of COVID-19 [29]. Yip’s study in the Hong Kong area found that the number of restaurants, public markets and clinics is positively related to the number of COVID-19 reported cases [30]. Yang proposed that the river-water environment may be a vector for viruses to transmit and pointed out that the risk of transmission in the water environment should not be ignored [31]. Studies have confirmed that the Yangtze River and Hanjiang River basins have higher transmission risk and different risk indexes. You pointed out that the incidence of COVID-19 was positively correlated with population density, proportion of construction land area, added value of unit land area of tertiary industry, density of public green space and density of elderly population and negatively correlated with average building scale and density of hospitals [32]. On this basis, some scholars have included social and economic factors in the selection and study of influencing factors. For example, Mollalo [19] explored the effect of socioeconomic and income inequality on the incidence of COVID-19 in their article. Scannell et al. also demonstrated that built environment characteristics in communities with high mortality rates from COVID-19 in the Chicago area include high floor area ratios [33]. There are some articles that are also exploring the relationship between socioeconomic factors and the distribution of morbidity [32]. In general, in the study of the transmission relationship between cities and COVID-19, the selection of relevant factors is mainly divided into four dimensions: urban open space factors, land use factors, urban facilities factors and social and economic factors.

Research on the current situation of urban geography reveals the relationship between health events and urban built environment and geographical space and points out issues that have long been neglected in planning urban security [34,35]. However, there are still some imperfect points in the research on this aspect. First of all, the research results on the correlation between built environmental factors and COVID-19 dispersal in existing literature are fragmented. Secondly, there are few empirical studies on the relationship between the built environmental factors and the risk of novel COVID-19 transmission from the perspective of space. Thirdly, most of the existing studies analyze the entire time period after the outbreak and lack studies for different time periods. Finally, few scholars have paid attention to the relationship between built environment and different users (e.g., different ages, different genders, etc.).

Our aim is to investigate the mechanisms by which the built environment influences the spatial distribution of different types of patients. In order to complete the study, the objectives of this paper are: first, to classify different populations; second, to investigate whether there are differences in the spatial distribution of different types of patients; third, to study the degree of influence of the built environment on the distribution of different types of patients; and finally, to analyze the potential mechanisms of influence, because according to the current situation, COVID-19 showed a characteristic of “elderly susceptible” in the early transmission period. There are differences in the proportion and number of infections in different populations. However, there are few analyses on this phenomenon in the existing studies.

To be able to draw conclusions, the research design of this paper is innovative based on previous research design. Since the composition and frequency of activities vary among different age groups, this paper divides them into young and middle-aged and elderly according to their most obvious characteristics, such as commuting behavior, based on realistic needs and existing studies. The study period is the natural transmission stage before the artificial control. During this period of time, people were not aware of the existence of the virus, had no control awareness or control measures for the virus; hence, the study of this period of time can explore the relationship between the built environment and the spread of the epidemic in its natural state of transmission. 

This paper investigates the correlation between the built environment and COVID-19 transmission. The results of the study not only help to reflect the characteristics of the relationship between the spread of the epidemic and the built environment in young and middle-aged and elderly populations but can also accurately put forward suggestions for the improvement of the urban built environment strategy so as to effectively control the spread of the epidemic and promote safe and healthy urban planning and management.

## 2. Materials and Methods

### 2.1. Study Area

With a resident population of 11.21 million, Wuhan has a large population base and high population density, making it a typical representative of high population density mega-cities in Asia. We select the core area within the third ring of Wuhan as the study area, covering seven administrative districts namely, Jiangan District, Jianghan District, Qiaokou District, Hanyang District, Qingshan District, Wuchang District and Hongshan District, with an area of about 860 square km. From the discovery of the first case to the control of the epidemic, the cumulative number of cases of COVID-19 has been 50,333. Wuhan saw China’’s first outbreak and has the largest number of cases of the megalopolises. During this period, the development of the epidemic was complete, which was conducive to the analysis and study of the natural transmission stage without human control. The results of this study have certain universality, can provide certain reference for epidemic prevention and control in other high-density cities in Asia.

### 2.2. Study Phase

The study period was from 1 December 2019, to 6 February 2020. This article focuses on the relationship between the built environment and the propagation of COVID-19 in the absence of human control before the closure of Wuhan City on 23 January 2020. The first COVID-19 case was found on 1 December 2019. Patients attacked by COVID-19 all infected before the closure of Wuhan city on 23 January 2020 (the incubation period of the virus is 14 days). Based on the spatial distribution of infected patients during this period, our study analyzed the correlation between built environment and COVID-19 transmission without active intervention, and further explored the correlation mechanism and action mechanism.

### 2.3. Data Source

The study data in this paper included the dependent variables—the spatial distribution of patients of different ages—and the independent variables—built environment factors. The population was divided into young and middle-aged people (aged between 18 and 59) and elderly people (aged over 60) based on the legal retirement age and differences in people’s outdoor activities in China. Gehl [36] divides outdoor activities into necessity activities and non-necessity activities that contain spontaneous activities and social activities. Necessary activities are activities that people are involved in to varying degrees, going to work, school, etc.; non-necessary activities include playing, talking, etc. In this article, patients are divided into young and middle-aged people and the elderly people through their behavior pattern. These two groups of people have significant differences in physiological functions, behavioral habits, psychological needs and other aspects, and the factors affecting the spread of the epidemic in the built environment are also different.

Since the cases visited and their basic information is non-public and difficult to obtain, we chose to use Sina Weibo help-seeking data as the dependent variable data. In the early days of the rapid spread of COVID-19’ in Wuhan, the supply capacity of medical resources was insufficient, and many confirmed cases could not be hospitalized. Sina Weibo, one of the most influential social media platforms in China, has opened an online help channel for COVID-19 patients by opening a network, which is also a government-approved help platform. Applicants are required to fill in their real name, age, city, address, contact information and a positive nucleic acid test result to ensure the authenticity of the information. Our study assumes that the distribution of medical resources in each district of Wuhan is fairly equal, and the overflow cases in each district are in an equal proportion to the actual cases. Total of 740 cases of public help-seeking data containing attributes such as age, address, and gender provided by the patient were obtained during this period, with effective data of 646 cases, including 286 middle-aged and young cases and 360 elderly cases. The sampling method for this study was similar to random sampling. To ensure the credibility of the study, all valid samples obtained from Wuhan city were used for the study to maximize the sampling proportion. As of 6 February 2020, the number of cases in Wuhan was 11,618, and the sample ratio in this study was about 5.5%.

The independent variable is the urban built environment factor. According to the existing literature research, the selection of variables focuses on three dimensions: land use, urban facilities and urban open space. The land use data (including building density and floor area ratio) is obtained through Baidu Maps with the community as the statistical unit. The urban facilities data (including commercial facilities, transportation stations and sports facilities, etc.) are obtained by point-of-interest (POI) crawling on Baidu Maps. The data of urban open space (including green space, water) comes from AMAP (Auto Navi Map).

### 2.4. Method

In order to investigate the mechanism and variability of built environment factors on the transmission of COVID-19 in young and middle-aged and elderly populations, the main urban area of Wuhan was used as the scope of this study and young and middle-aged and elderly patients in Wuhan were the study subjects. Using Sina Weibo help-seeking data, 14 urban built environment factors under three dimensions of land use, urban open space, and urban facilities were selected, and multiple linear regression models were used to explore the key built environment factors influencing the risk of COVID-19 infection in middle-aged and older populations, and to compare the similarities and differences between the two models.

The specific steps of the study are shown in the figure below (Figure 1). The study made a clear division of young and middle-aged and elderly people, which determines the selection of dependent variable of built environment. The study also removed the multicollinearity of numerous factors. Multiple linear regression was performed between the spatial distribution of patients and the built environment factors of the two age groups, and the significant correlation of the influencing factors between the two groups was obtained and contrastively analyzed.

#### 2.4.1. OLS (Ordinary Least Squares) Estimation in Multivariate Linear Regression

Multiple linear regressions normalized the built environmental factors in the study, and then carried out linear regression. The regression coefficient obtained reflected the importance of the influence of the corresponding urban built environmental factors on the community distribution of COVID-19 patients. At this time, the regression equation [37] is called the standard regression equation, and the regression coefficient is called the standard regression coefficient, which can be expressed as follows:(1)$$y=\beta_{0}+\beta_{1}x_{1}+{\;\beta }_{2}x_{2}+\cdots +\beta_{i}x_{i}+\varepsilon$$
where $\beta_{i}$ represents the regression coefficient of the i-th factor and $x_{i}$ represents the independent variable used in the article, including several factors, such as building density, floor area ratio, and college density; y represents the two dependent variables in the two models, i.e., kernel density for Sina Weibo data seeking help within young and middle-aged and kernel density for Sina Weibo data seeking help within elderly, respectively.

#### 2.4.2. Kernel Density Analysis

Kernel density analysis is used to calculate the unit density of urban facility elements within the specified neighborhood. In this paper, the central urban area of Wuhan is divided into 1062 districts by communities. The kernel density analysis of facilities is performed for each point in the map, and the number of POI points of various urban facilities within the range is counted according to the service radius of different facilities. Then the average value of the kernel density statistics in each community was taken as the unit to quantify the number of facilities to each community.

Regarding the cell size for the kernel density analysis, a cell size of 500 m × 500 m is chosen for the result output in this study. According to the Chinese urban planning code, the average urban road spacing is about 500 m. Therefore, we believe that this size is similar to the average size of community units and can show the community situation more clearly. Regarding the determination of the search radius when conducting the kernel density analysis, it is determined according to the service radius of each type of facility (500–3000 m). For example, in primary school facilities, 500 m was chosen as the search radius for China’s regulations regarding their service area. As for the secondary school facility, 1500 m was chosen as its search radius according to the service radius.

#### 2.4.3. Near Analysis

Near analysis is used to calculate the distance of the nearest arc, point, or node from each point in the element set to another. For example, the distribution of water system in urban open space is relatively concentrated, with large area and long boundary, so it is not suitable to use kernel density for analysis. Near analysis will identify the nearest distance of the community to the water system. In order to improve the feasibility of subsequent regression analysis, some of the faceted data and the less quantified point data in the independent variables were linked to the dependent variables.

### 2.5. Determination and Treatment of Variables

#### 2.5.1. Dependent Variable

The first step of dependent variable processing is to collect the information on the age and address in the help-seeking data and calculate the kernel density of the patients. Then—to integrate unit data by community, the value of each community being the average value of the unit point’’s kernel density. The final step is to get spatial distribution of the help kernel density of the middle-aged and young and the elderly (Figure 2).

#### 2.5.2. Independent Variable

The independent variables focus on three dimensions: land use, urban open space and urban facilities. Under the premise of equal medical resources, this study first excluded the medical related factors in the dimension of urban facilities. The grade and service radius of different urban facilities are different, which will have a great impact and difference on ’people’s use frequency and cost. According to the service radius, urban facilities are divided into district level and community level. Kernel density and near analysis were used. In order to exclude multicollinearity of factors, the least square method (OLS regression analysis) was used to calculate the VIF (variance inflation factor) value (variance expansion coefficient) of the factors. The high multicollinearity factor over five was eliminated, and 14 selected factors were finally used for multiple linear regression. (Table 1)

## 3. Result

### 3.1. Results of Analysis on Young and Middle-Aged Persons

Each factor is represented by its kernel density. As shown in Table 2, the OLS model analysis results of young and middle-aged people seeking help from Weibo showed that the bus stations, the middle schools and the floor area ratio were positively correlated with the kernel density distribution of help-seeking. However, ‘universities and community sports facilities are negatively correlated with the distribution of the kernel density of help-seeking (Figure 3). At the same time, the adjusted R square of the regression model for young and middle aged is 0.320 (Table 3), which indicates that the 14 selected independent variables are reliable for the dependent variables.

### 3.2. Results of Analysis on Elderly

The results of the OLS model analysis of the elderly Weibo users seeking help (Table 4) are shown, bus station, community-level shopping facilities, building density, middle school, community-level sports facilities, community-level water system, community-level sports facilities and district-level cultural facilities were all positively correlated with the kernel density distribution of help-seeking. Universities, community-level catering facilities and community distance from green space were negatively correlated with the kernel density distribution of help-seeking (Figure 4). The adjusted R squared of the regression model for the elderly was 0.613 (Table 5), indicating that the explanatory power of the selected 14 independent variables to the dependent variables was credible.

### 3.3. Results Contrast

Compared with the regression analysis results (Figure 5) of the two models, there were four common significant influencing factors in the two age groups: bus Station, middle school, university, community sports facility. There were six more factors related to the elderly than the young and middle aged: building density, community-level shopping facilities, community-level catering facilities, community distance from water system, community distance from green space, community distance from district-level sports facilities, compared with the elderly, young and middle-aged people have more volume rate of this one significant factor. With the increase of age, more factors of urban facilities were significant.

This result is the same as that of scholars Yip [30], who mentioned in their article that three built environmental factors, public transportation, restaurants and public markets, are all positively influencing the prevalence of COVID-19 at different stages.

## 4. Discussion

### 4.1. Spatial Distribution of Patients

There is a great difference in spatial distribution between elderly patients and young and middle-aged patients. The distribution of elderly patients has obvious aggregation. They are mainly concentrated in Zhongshan Park in Hankou District, Mayinglu Wetland in Hanyang District, Shahu Park in Wuchang District and Qingshan Park in Qingshan District. The above areas are all located around large public open space, with relatively dense population distribution and high density of elderly people. The distribution of young and middle-aged patients did not have obvious aggregation but was more discrete. According to the distribution of the whole city, the incidence density in Hankou area is higher than that in other areas. This area contains the main types of business districts, and the public transportation stations are complete and dense. The entertainment and work behaviors of young and middle-aged people mostly occur in this area (Figure 2).

COVID-19 is a respiratory infectious disease that is associated with daily activity patterns. Here, we describe that the elderly has less cross-regional activities, little spatial displacement, and strong spatial agglomeration after infection. Young and middle-aged people mainly carry out necessary commuting activities, and the proportion of cross-district commuting is high, so the spatial distribution of patients will show a high dispersion.

Scannell [33] mentioned in the article the study of population classification. The finding that “52% of deaths among white residents occurred in nursing homes” is similar to the finding in this article that “the distribution of elderly patients was clustered”.

### 4.2. Analysis of Linear Regression Results of Kernel Density for Sina Weibo Data Seeking Help within Young and Middle-Aged

#### 4.2.1. Land Use

The floor area ratio has a positive correlation effect on the distribution of the young and the middle-aged seeking help, the larger the floor area ratio is, the greater the seeking help young and middle-aged people in the community. The plot ratio of Sanshu community is 3.83, while that of Zhengjie community is only 1.76. Young and middle-aged people in Sanshu community are three times that of Zhengjie community. To some extent, this indicates that communities with a larger floor area ratio have a relatively dense population, which will increase the incidence density and promote the spread of the epidemic.

#### 4.2.2. Urban Facilities

Bus stops and middle schools are positively correlated with the density of young and middle-aged people seeking help [27]. These two types of areas with dense urban facilities usually have high accessibility and high degree of population mobility, and the distribution of large population will provide favorable conditions for the spread of COVID-19. This is similar to the results of Scarpone’s study [38]. The article says, “Transportation Stations serve as nodes where high densities of travelling persons increase the probability intra-population contagion.”

University and community sports facilities are negatively correlated with young and middle-aged people seeking help. Wuhan’s universities have clear boundaries and are often managed in a closed way. The dense area of colleges and universities will hinder the communication and contact of people, and it will restrain the spread of the epidemic to a certain extent. Physical fitness is good per capita for young and middle-aged people where community-level sports facility density is great, so they are not easily infected by COVID-19, leading to a low incidence of the disease.

### 4.3. Analysis of Linear Regression Results of Kernel Density for Sina Weibo Data Seeking Help within Elderly

#### 4.3.1. Land Use

Building density has a positive correlation effect on the distribution of the elderlies seeking help. In other words, the higher the building density, the higher the elderlies seeking help. The building density of Hongshanfang community in Wuchang district is 0.24, and the building density of Daqiao’s first community in Hanyang district is only 0.17. The former is about five times that of the latter. The quality of ventilation conditions is an important condition for the spread of the epidemic of COVID-19 [39]. The community with high building density usually has poor ventilation, which is easy to accelerate the spread of the epidemic.

#### 4.3.2. Urban Facilities

Elderly patients show a high correlation with the factors of urban facilities. Bus stations, community-level shopping facilities, middle schools and community-level sports facilities are all positively correlated with elderly patients’ seeking help. According to the reality, the use of such facilities often promotes spatial contact between people, leading to high aggregation and increasing the risk of infection. For example, the Jianghan Road Business District in Wuhan is one of the areas with the most intensive shopping facilities and public transportation facilities in Wuhan, and the release rate of help information for elderly patients in this node is relatively high. There was a negative correlation between college and elderly patients. Colleges and universities in Wuhan have a clear scope and boundary, the relative isolation between the campus environment and the outside world will hinder the communication of people, which has a certain inhibition effect on the spread of the epidemic.

#### 4.3.3. Open Space

The factor of urban open space has a significant impact on elderly patients’ help-seeking. There is a negative correlation between the distance of green space and the elderly, while the distance of water system was completely opposite. It is a conjecture that communities closer to public green spaces, the frequency of elderly people going to green spaces to carry out public activities will increase, and the chances of contact with people will increase, leading to the increased probability of infection. The highest density of elderly seeking help was distributed around the parks of each district. This result and opinion are the same as that of researchers such as You [32]. They believe that high public green space density increases outdoor activity opportunities [32]. Therefore, high public green space density increases the COVID-19 morbidity rate. However, in communities far from water system, the space is more open, and the community ventilation is better, so the probability of infection is lower.

### 4.4. Comparative Analysis of Influence Factors between Young and Middle-Aged Patients and Elderly Patients

Compared with young and middle-aged people, old people had more significant influence factors such as building density, community-level shopping facilities, community-level catering facilities, community distance from water system, community distance from green space and community distance from district-level sports facilities. The community-level shopping facilities are significant in the elderly population probably because the elderly cannot use the network skillfully and depend on the physical shopping facilities [40]. Urban open space factors are significant among elderly patients perhaps because the attributes of the community are more influential for the elderly because they spend a lot of time in the community doing non-essential activities. Young and middle-aged people mainly engage in necessary commuting activities, and the environmental quality of the community cannot be a significant factor affecting their infection.

The common significant influencing factors in the two age models are bus stations, middle school, universities and community sports facilities. However, each factor has different regression coefficient in the two models, and the same factor has different influence on the two age groups.

The effect of bus stop is more significant for young and middle-aged people, perhaps because they need to perform necessary commuting activities, are more dependent on public transportation and use it regularly. The isolation of the university environment from the outside world has a negative effect on the spread of the epidemic. The absolute value of the coefficient after regression analysis is higher for the elderly patients than for the young and middle-aged patients. Young and middle-aged people communicate and interact more with colleges and universities due to their study and work needs, and colleges and universities do not separate their activities as much as the elderly. The final analysis results can also better explain this phenomenon. Community sports facilities has a completely opposite effect on the two groups of people, which is positively correlated with the elderly and negatively correlated with the young and middle-aged people. We speculate that this might be due to the large differences in the physical qualities of these two groups. For young and middle-aged people, the high density of sports facilities may improve the physical fitness of young and middle-aged people and make them less prone to infection with the virus. For the elderly, although the use of sports facilities can play a role in improving physical fitness, the immunity is still generally lower than the young, and the use of sports facilities will cause gathering, the risk of infection of the virus will be greatly increased.

### 4.5. Suggestions

Floor area ratio and building density showed significant influence in each of the two categories of patients. Therefore, at the level of land development and utilization, appropriate capacity control should be carried out for high-density residential areas with high floor area ratio. To ensure ventilation and regular disinfection of the surrounding buildings and public spaces, the development and construction of high-rise buildings should be carried out cautiously.

According to our speculation, the inability of the elderly to use online facilities proficiently makes elements of urban facilities an important medium for the spread of the virus among the elderly. So, we have some suggestions for improving the urban facilities. On the basis of ensuring complete facilities, online service functions of all kinds of facilities should be developed to facilitate people’s daily use, reducing queuing and gathering time for offline use of facilities, and reducing the transmission of infectious diseases. Since elderly patients show a stronger significant correlation to various facilities, the needs of the elderly with low immunity should be considered more when considering the layout and use of facilities, and help the elderly learn how to use online facilities. According to the results of the study, transportation facilities showed more significance for middle-aged and young people, so the layout of facilities related to daily commuting should be paid more attention. It is suggested to travel at different peak times and use private transport instead of public transport. Transportation facilities play a great role in promoting the cross-regional transmission of the epidemic, so the urban layout mode of TOD (transit-oriented development) should be carefully considered and optimized. At the same time, regular ventilation and disinfection should be strengthened for traffic facilities.

According to the results of the study, both urban green spaces and water areas have a strong influence on geriatric patients. Thus, in the construction of urban open space, attention should be paid to the building of urban air corridor to improve urban air quality. The active construction of public green space in communities emphasizes social equity, provides more choices for community residents, and avoids the centralization of urban green space, which will aggravate the cross-infection of communities in a wide range. Large green spaces should also be regularly disinfected. For cities with developed water system, it can be considered to combine water system and large green space layout and reduce the infection risk of public open space by using urban wind corridors.

### 4.6. Study Limitations

There are still some limitations in revealing the mechanism of the association between the built environment and the risk of communicable diseases among young and middle-aged people and the elderly.

First of all, the analysis and processing of the case data for Weibo help did not consider the individual differences in the samples. This study did not consider the specific work situation of different individuals aged around 60. We only idealized the age of 60 as a cut-off for commuting behavior. This may lead to studies that do not yield completely accurate results.

Secondly, there are some MAUP (modifiable areal unit problem) in this study, mainly about the scale effect in the MAUP. This paper starts the study with the community as the unit, and the values of each variable are averaged within the community, i.e., the area data with the community as the statistical unit. Since the size of each community in Wuhan varies, the precision of the area data with the community as the statistical unit varies. Smaller communities have higher precision, while larger communities will have lower precision. This will have a slight impact on the precision of the study results, while the explanatory power of the model for larger communities will also be somewhat affected. In future studies where the precise location of cases can be obtained, this may be improved by decomposing the larger communities to count them again, trying to make all study units the same size. The impact of MAUP on the results of spatial statistical analysis can also be minimized by performing integrated analysis under multiple scale schemes.

Thirdly, the mechanism behind different influencing factors is only theoretical research, and further demonstration is needed. At the same time this is only a speculative mechanism of action based on data from Wuhan city, perhaps these mechanisms will vary in different regions.

Finally, this study takes Wuhan as an example, and the research results are only applicable to cities with high population density and lack guidance for small and medium-sized cities with low population density.

## 5. Conclusions

This study focused on the time period before the human-controlled epidemic and analyzed the relationship between the middle-aged and elderly populations influenced by built environment factors during the spread of COVID-19, using 740 cases of Sina Weibo help-seeking data in the main city of Wuhan, and conducted a differential study on the influence of built environment on the spread of the epidemic on middle-aged and elderly people. The results showed that, first, the built environment factors were significantly related to the transmission risk of infectious diseases. Second, there was a great difference in the spatial distribution between young and middle-aged patients and elderly patients. Third, the spread of COVID-19 in the two age groups was affected by the built environmental factors in different types and degrees.

In the elderly population, the virus transmission is more easily affected by the environment and there are more built environment factors with significant influence than in the young and middle-aged population. More urban facilities factors show significance in the elderly population. In the future of urban construction, we should not only make use of the open space and water system of the city, but also pay attention to the urban ventilation to ensure the good sanitation condition of the city. The biggest impact among the young and middle-aged groups comes from the commuting facilities. Therefore, in the future, in urban construction attention should be paid to commuting facilities to keep them clean and disinfected regularly.

In this study, some different mechanisms of virus transmission in the built environment to young and middle-aged and elderly populations were derived, which may provide a reference for the prevention and control of major airborne public health events in the future.

## Figures and Tables

**Figure 1 ijerph-18-07563-f001:**
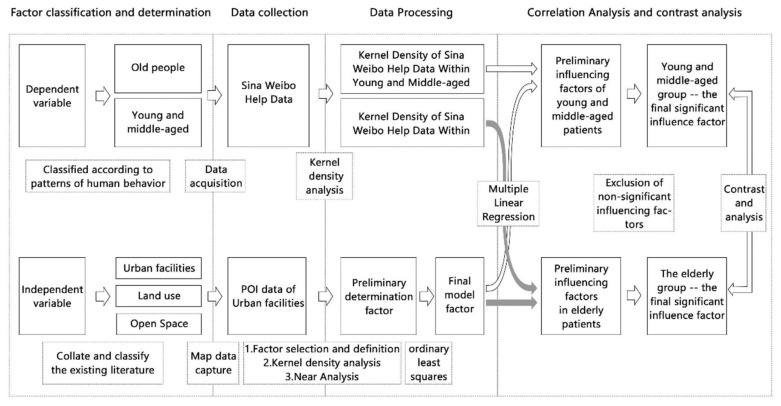
Schematic diagram of the experiment.

**Figure 2 ijerph-18-07563-f002:**
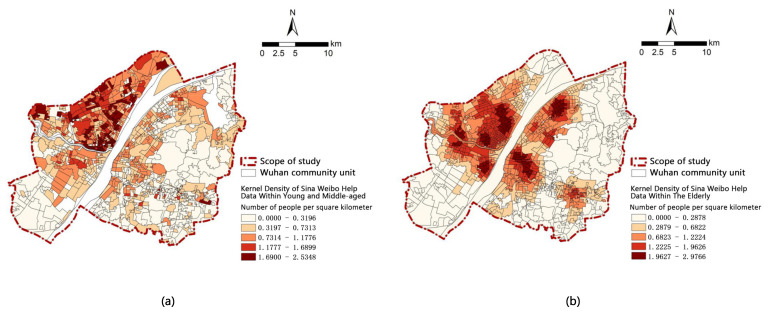
Kernel density spatial distribution of Sina Weibo help data: (**a**) kernel density spatial distribution of Sina Weibo help data within young and the middle-aged; (**b**) kernel density spatial distribution of Sina Weibo help data within the elderly.

**Figure 3 ijerph-18-07563-f003:**
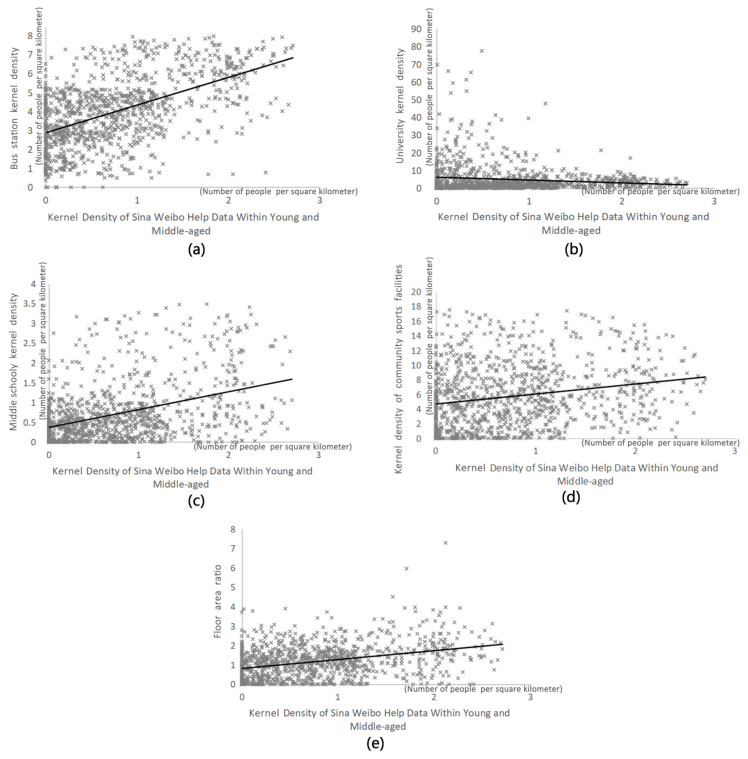
Scatter plot of significant influence factor within young and the middle-aged: (**a**) bus station kernel density; (**b**) college kernel density; (**c**) secondary kernel density; (**d**) kernel density of community sports facilities; (**e**) floor area ratio.

**Figure 4 ijerph-18-07563-f004:**
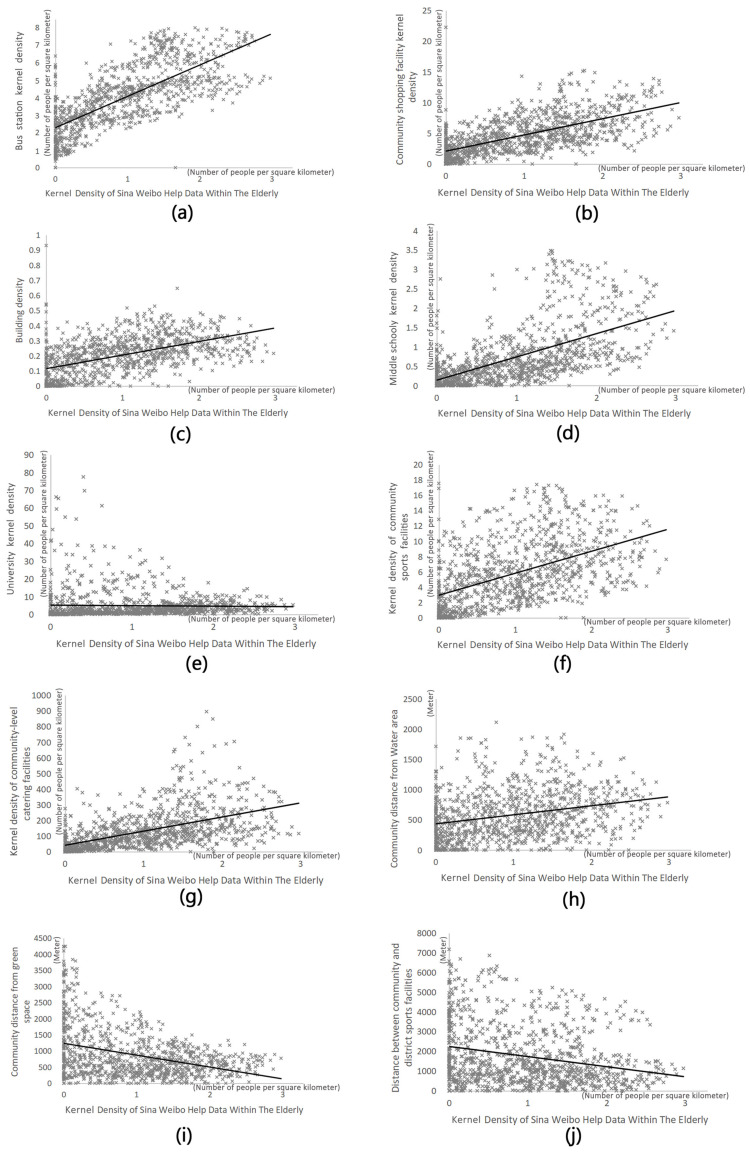
Scatter plot of significant influence factor for the elderly: (**a**) bus station kernel density; (**b**) community shopping facility kernel density; (**c**) building density; (**d**) secondary kernel density; (**e**) college kernel density; (**f**) kernel density of community sports facilities; (**g**) kernel density of community-level catering facilities; (**h**) community distance from water area; (**i**) community distance from green spaces; (**j**) distance between community and district sports facilities.

**Figure 5 ijerph-18-07563-f005:**
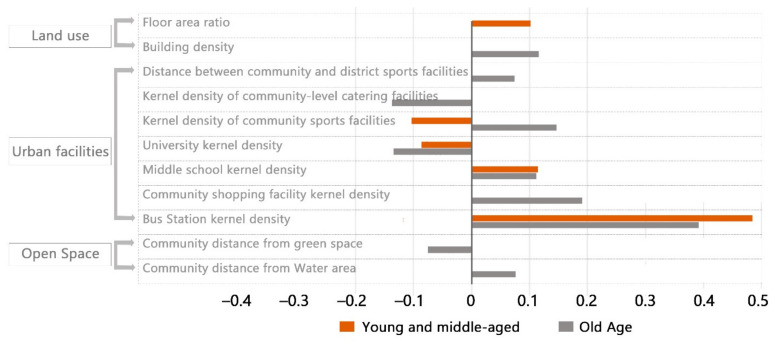
Built environment factor influence coefficient.

**Table 1 ijerph-18-07563-t001:** The category, name and calculation method of impact factor.

Factor Category		Factor Name	Factor Calculation Method
Land use		Building density	The ratio of the total base area of a building to the occupied area within a certain range (%)
		Floor area ratio	The ratio of the total above-ground floor area to the net land area of a community
Urban facilities	Educational facilities	Kindergarten and primary school kernel density	Refers to kindergartens and primary schools in the central district of Wuhan. The calculation radius of kernel density is 500 m, taking the average of each point in the community
		Middle school kernel density	A middle school in central Wuhan with a kernel density calculation radius of 1500 m, taking the average of each point in the community
		University kernel density	Refers to institutions of higher learning in the central district of Wuhan. The calculated kernel density has a radius of 1500 m, taking the average of each point in the community
	Cultural facilities	Kernel density of cultural facilities at district level	Including libraries, museums and other facilities, kernel density calculation search radius of 3000 m, take the community point average
	Catering facilities	Kernel density of community-level catering facilities	Refers to Chinese restaurants, fast food restaurants and restaurants in the central district of Wuhan. The calculated kernel density has a radius of 600 m, about a five-minute walking distance, and is taken as the average of all points in the community
	Shopping facilities	Kernel density of community shopping facilities	The kernel density of supermarkets, convenience stores and so on in Wuhan’s central district is calculated with a search radius of 1000 m, taking the average of all points in the community
	Sports facilities	Kernel density of community sports facilities	Wuhan sports venues, gyms, sports facilities and so on, kernel density calculation search radius of 1500 m, take the average of the community points
		Distance between community and district sports facilities	The sports facilities are primarily Wuhan coliseum complexes, and since the university coliseum is not entirely open to the public, the study excluded the university’s internal coliseum when selecting POI sites. This indicator describes the straight-line distance from each community center to the nearest gymnasium
		Kernel density of recreational facilities at district level	It mainly includes cinemas, amusement parks and other facilities in the central area of Wuhan. The kernel density is calculated with a search radius of 3000 m, taking the average of all points in the community
		Kernel density of Bus Station	Taking the community as the unit of calculation, the average value of the bus station kernel density of each point in the community was calculated
Open space		The distance of the community from the water	The shortest distance from the center of the community to the boundary of the water
		The distance between the community and the green space	The shortest distance from the center of a community to the boundary of a green space

**Table 2 ijerph-18-07563-t002:** Result of the OLS model for the young and middle-aged.

Model	Nonstandard Coefficient		Normalized Coefficient	t	Significance
	B	Standard Error	Beta		
(constant)	−0.018	0.054		−0.331	0.741
Kernel density of bus station	0.169	0.013	0.486 **	12.523	0.000
University kernel density	−0.007	0.002	−0.086 *	−2.829	0.005
Middle school kernel density	0.095	0.029	0.114 **	3.284	0.001
Kernel density of community sports facilities	−0.015	0.006	−0.103 *	−2.669	0.008
Floor area ratio	0.072	0.023	0.102 **	3.138	0.002
The distance between the community and the green space	0.000	0.000	0.061	2.209	0.027

Note: dependent variable: the kernel density of young and middle-aged seeking help on Weibo; ** indicated that it was able to pass the statistical test with a significance level of 0.5%; * indicated that it was able to pass the statistical test with a significance level of 1%.

**Table 3 ijerph-18-07563-t003:** Summary of the OLS model for young and middle-aged.

R	R-Square	Adjusted R-Square	Error in Standard Estimation
0.569	0.324	0.320	0.518

**Table 4 ijerph-18-07563-t004:** Result of the OLS model for elderly.

Model	Nonstandard Coefficient		Normalized Coefficient	t	Significance
	B	Standard Error	Beta		
(constant)	−0.237	0.059		−4.050	0.000
Kernel density of bus station	0.162	0.013	0.394 **	12.047	0.000
Community shopping facility kernel density	0.045	0.008	0.191 **	5.671	0.000
Building density	0.685	0.157	0.116 **	4.360	0.000
Middle school kernel density	0.111	0.032	0.112 **	3.524	0.000
University kernel density	−0.012	0.002	−0.134 **	−5.396	0.000
Kernel density of community sports facilities	0.026	0.006	0.147 **	4.425	0.000
Kernel density of community-level catering facilities	−0.001	0.000	−0.137 **	−3.743	0.000
The distance of the community from the water	0.000	0.000	0.076 **	3.641	0.000
The distance between the community and the green space	−0.000	0.000	−0.075 **	−3.445	0.001
Distance between community and district sports facilities	0.000	0.000	0.074 **	3.254	0.001
Kernel density of cultural facilities at district level	0.021	0.009	0.080	2.216	0.027

Note: dependent variable: the kernel density of young and the middle-aged seeking help on Weibo; ** indicated that it was able to pass the statistical test with a significance level of 0.5%; * indicated that it was able to pass the statistical test with a significance level of 1%.

**Table 5 ijerph-18-07563-t005:** Summary of the OLS model for elderly.

R	R-Square	Adjusted R-Square	Error in Standard Estimation
0.785	0.617	0.613	0.464

## Data Availability

Restrictions apply to the availability of these data. The Weibo Help data was obtained from the Sina Weibo application and is available from the Sina Weibo website (https://weibo.com, accessed on 2 July 2021) with the permission of the Sina Weibo platform. POI for city facilities were obtained from the Amap application and are available from the Amap website (https://www.amap.com, accessed on 2 July 2021) with the permission of Amap.

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
