# Peer review of "Relationship between Built Environment and COVID-19 Dispersal Based on Age Stratification: A Case Study of Wuhan"

_ijerph, 2021, doi:10.3390/ijerph18147563_

Round 1

Reviewer 1 Report

Dear authors,

Your manuscript is interesting but I need you to answer some questions:

MATERIALS AND METHODS

Study Phase:

  • The authors must specify the research design.
  • The authors provide redundant information (“Introduction”). You must simplify the information.
  • How was the sample chosen? You must specify how the sampling was done.

Kernel density analysis:

  • Regarding the cell size for the kernel density analysis, Is the chosen size representative of the rest of the city?

REFERENCES

  • Some references are incomplete or have errors. The authors should review this section.

Author Response

Dear editor:

On behalf of my co-authors, we thank you very much for giving us an opportunity to revise our manuscript, we appreciate editor and reviewers very much for their positive and constructive comments and suggestions on our manuscript entitled “Relationship between built environment and COVID-19 dispersal based on age stratification:A case Study of Wuhan”. (ID: ijerph-1248641)

Those comments are all valuable and very helpful for revising and improving our paper, as well as the important guiding significance to our researches. We have studied comments carefully and have made correction which we hope meet with approval. We have use the "track changes" feature in Microsoft Word. This makes it easier for editors and reviewers to see the changes that have been made the main corrections in the paper and the responds to the reviewer’s comments are as flowing:

Reviewing: 1

1.Comment: The authors must specify the research design.

Response to comment: Thank you for your suggestion regarding the study design, which we have added in Section 2.4, and the specific study steps are shown mainly through Figure 1.

2.Comment: The authors provide redundant information (Introduction"). You must simplify the information.

Response to comment: Thank you for your suggestion, we have streamlined the introductory section without affecting the general meaning of the paragraph.

3.Comment: How was the sample chosen? You must specify how the sampling was done.

Response to comment: Thank you very much for pointing out our shortcomings! The selection about the sample and the sampling proportion are described in more detail in section 2.3. Regarding the issue of sampling methods, they have been added in the same section based on the reviewers' comments.

4.Comment: Kernel density analysis: Regarding the cell size for the kernel density analysis, ls the chosen size representative of the rest of the city?

Response to comment: Thank you for your question! Yes, since the cell size was determined by us based on the average size of the Wuhan community, the selected size is representative of the rest of the city.

5.Comment: Some references are incomplete or have errors. The authors should review this section.

Response to comment: We apologize for the error about the reference; we have checked it again to make sure it is correct. Some reference I'm not sure how to abbreviate the journal title, so leave the entire title.

With the reviewers' suggestions, we improved the manuscript and made some changes to the manuscript. These changes will not influence the content and framework of the paper. And here we did not list the changes but marked in red in revised paper. We appreciate for Editors/Reviewers’ warm work earnestly and hope that the correction will meet with approval.

Thank you and best regards

Yours sincerely,

Corresponding author: Jie Shen

Reviewer 2 Report

Thank you for amending your manuscript. I believe this updated version is much improved.

However, I'm afraid the text is still a little difficult to follow in places and the manuscript would benefit from a thorough editing of English language.

Also, in the Results there are several instances that read "...correlated with the kernel density distribution". I think these would be clearer if written as "...correlated with help-seeking".

Author Response

Dear editor:

On behalf of my co-authors, we thank you very much for giving us an opportunity to revise our manuscript, we appreciate editor and reviewers very much for their positive and constructive comments and suggestions on our manuscript entitled “Relationship between built environment and COVID-19 dispersal based on age stratification:A case Study of Wuhan”. (ID: ijerph-1248641)

Those comments are all valuable and very helpful for revising and improving our paper, as well as the important guiding significance to our researches. We have studied comments carefully and have made correction which we hope meet with approval. We have use the "track changes" feature in Microsoft Word. This makes it easier for editors and reviewers to see the changes that have been made the main corrections in the paper and the responds to the reviewer’s comments are as flowing:

Reviewing: 2

1.Comment: I'm afraid the text is still a little difficult to follow in places and the manuscript would benefit from a thorough editing of English language.

Response to comment: Based on the reviewers' suggestions, we made every effort to adjust the wording of the text to ensure that it was easy to understand with clarity of expression.

2.Comment: In the Results there are several instances that read “...correlated with the kernel density distribution”. I think these would be clearer if written as “...correlated with help-seeking”.

Response to comment: Thank you for your suggestion, we have adjusted our wording.

With the reviewers' suggestions, we improved the manuscript and made some changes to the manuscript. These changes will not influence the content and framework of the paper. And here we did not list the changes but marked in red in revised paper. We appreciate for Editors/Reviewers’ warm work earnestly and hope that the correction will meet with approval.

Thank you and best regards

Yours sincerely,

Corresponding author: Jie Shen

Reviewer 3 Report

This is an interesting paper describes the probable relationship between COVID-19 dispersal and built environment in an example city. The paper is well presented in general. The paper is recommended for publication with a moderate revision.

Abstract. Please define the age range for the young and middle-aged and elderly patients.

Section 2.4.1, the last paragraph, misprinted line. The equation should be numbered and described in the text. Please give references for the equation.

Please link up the equation variables to the list (Table 1) for explanatory variables and factors; please also give units for the variables. It is about a matter of concern in the generality of the equation application. Floor area was imposed in the selected factors and reflected, to some extent, by the kernel density. Plot ratio was included in the discussion section, however, it was not clearly mentioned in the methodology nor those for the selected factors. Please give more information why various radius was selected for the factors (600-3000 square meter)? Please give information about the floor area ratio of the facilities. It would be a useful factor in town planning.

Figure 5 should be explained in the text (Section 3.3?).

Figures. Legend, axis labels are too small to be recognized. Variables in axis should be presented with units.

Table 3 & others. Please give suitable decimal places for error in standard estimates.

Numeric values of ten or below presented in English are suggested, e.g. one for 1, four for 4, ten for 10, etc.  

Author Response

Dear editor:

On behalf of my co-authors, we thank you very much for giving us an opportunity to revise our manuscript, we appreciate editor and reviewers very much for their positive and constructive comments and suggestions on our manuscript entitled “Relationship between built environment and COVID-19 dispersal based on age stratification:A case Study of Wuhan”. (ID: ijerph-1248641)

Those comments are all valuable and very helpful for revising and improving our paper, as well as the important guiding significance to our researches. We have studied comments carefully and have made correction which we hope meet with approval. We have use the "track changes" feature in Microsoft Word. This makes it easier for editors and reviewers to see the changes that have been made the main corrections in the paper and the responds to the reviewer’s comments are as flowing:

Reviewing: 3

1.Comment: In abstract, please define the age range for the young and middle-aged and elderly patients.

Response to comment: Based on the reviewers' comments, we have added a note on age in the abstract for ease of reading. And the detailed description of the age stratification is given later.

2.Comment: In Section 2.4.1, the last paragraph, misprinted line. The equation should be numbered and described in the text. Please give references for the equation. Please link up the equation variables to the list (Table 1) for explanatory variables and factors; please also give units for the variables. It is about a matter of concern in the generality of the equation application.

Response to comment: Based on the reviewer's comments, we corrected the error in the last paragraph of section 2.4.1 and numbered the equations and refined the descriptions to relate the variables in the equations to the actual factors in this study. We also added references to the equations. Regarding the units of the variables, we have refined the units of variables in Section 3.1 as well as in the figures in Section 3.2, so that the units of each relevant variable can be clearly seen.

3.Comment: Floor area was imposed in the selected factors and reflected, to some extent, by the kernel density. Plot ratio was included in the discussion section; however, it was not clearly mentioned in the methodology or those for the selected factors. Please give more information why various radiuses were selected for the factors (600-3000 square meter)? Please give information about the floor area ratio of the facilities. It would be a useful factor in town planning.

Response to comment: We thank the reviewers for their comments. The volume ratios were selected by studying the existing literature. And due to our oversight, they were not shown in the introduction section. Now we have improved the literature review related to the selection of floor area ratio. As for the search radius in the kernel density analysis, we give a more detailed explanation in section 2.4.2, which is based on the service radius of each facility in China.

4.Comment: Figure 5 should be explained in the text (Section 3.3?).

Response to comment: Thanks to the reviewer's comments, chapter 3.3 is mainly about the objective description of Figure 5. And further analysis and interpretation is placed in the discussion in chapter 4.

5.Comment: Legend, axis labels are too small to be recognized. Variables in axis should be presented with units.

Response to comment: Thanks to the reviewers for their corrections! Based on the reviewers' suggestions, we have enlarged the pictures, legends and axis labels, and filled the units of the variables.

6.Comment: Table 3 & others. Please give suitable decimal places for error in standard estimates.

Response to comment: We thank the reviewers for their corrections, and we have corrected the decimal places for the table based on the reviewers' comments.

7.Comment: Numeric values of ten or below presented in English are suggested, e.g. one for 1, four for 4, ten for 10, etc.

Response to comment: Based on the reviewers' comments, we have revised this presentation accordingly.

With the reviewers' suggestions, we improved the manuscript and made some changes to the manuscript. These changes will not influence the content and framework of the paper. And here we did not list the changes but marked in red in revised paper. We appreciate for Editors/Reviewers’ warm work earnestly and hope that the correction will meet with approval.

Thank you and best regards

Yours sincerely,

Corresponding author: Jie Shen

Round 2

Reviewer 1 Report

Dear authors,

Thanks for your reply. The explanations of the authors are satisfactory. The paper has greatly improved its quality.

Congratulations on your work.

Best regards

Author Response

Dear reviewer:

Thank you for your professional advice and hard work, which has helped us a lot to improve the quality of our article.

Thanks again for good comments and kind considerations.

Best regards

This manuscript is a resubmission of an earlier submission. The following is a list of the peer review reports and author responses from that submission.

Round 1

Reviewer 1 Report

This paper used the COVID-19 patient data from a social media, Weibo, to investigate the impacts of the built environment on different age groups. However, this paper lacks academic writing standards, and the research did not have scientific rigorousness. Therefore, rejection is recommended. Below are detailed comments. 1. The academic writing of the paper needs improvement. For example, references are missing in the first paragraph, which could be considered plagiarism. 2. Method needs to be further clarified. a. The dependent variable was stated as "the spatial distribution of patients of different ages." It is not clear and confusing how the dependent variable is measured. What is the spatial unit of the dependent variable? b. kernel density analysis was used in different built environmental measures as well as the dependent variable. However, it is unclear what cell size and search radius were used when doing kernel density analysis. c. There is a lack of theory on how different built environment attributes were selected.

Reviewer 2 Report

I have carefully considered and read the manuscript entitled “Relationship between built environment and COVID-19 dispersal based on age stratificationA case Study of Wuhan” and have the following observations:

The outbreak of COVID-19 has become the focus of attention in the field of urban geography. Built environment is an important factor affecting the spread of the epidemic. However, different activity patterns in different age groups have different relationships with the built environment. Based on this, our study conducted a regression analysis of the spatial distribution of patients and the built environment based on the classification of human behavior patterns in Wuhan city, which was not controlled by human intervention. The results showed that the spatial distribution of young and middle-aged patients was more homogeneous, while the spatial distribution of elderly patients had a strong clustering characteristic; Moreover, the significant built environment factors showed great differences between the two groups.

Major comments and Suggestions for Authors:

This paper is not with enough clarity about the aims and objectives, so please do it to clarify for more understanding to meet the standard of readership of IJERPH. There are some errors in spelling, and some more clarifications, improvements in modeling techniques, acute conclusion, policy recommendations, and research limitations are needed for reconsidering that manuscript for publication in the International Journal of Environmental Research and Public Health.

In addition to the above, I have a few points for the authors to consider before the publication of this work:

  • The abstract should check thoroughly and compose with a summative style without spelling mistakes and more focused on the main impacting results and policy implications.
  • Please highlight your contribution and novelty of this manuscript with accuracy in the introduction part before the arrangement description. Furthermore, the objectives of your study should elaborate clearly there in the introduction part.
  • The literature and theoretical background and hypothesis construction should improve and add more relevant studies e.g. (latest) to grab and display more contemporary literature critically.
  • Please update your literature with few latest studies if it is suitable and improve its style as well.
  • Recheck the references and their style is according to the journal requirements, and in-text and end-text should be the same and vice versa.
  • In the Methodology part, please more detailed the results and discussion in presence of constructed hypothesis part for the actual output of this study for stakeholders and targeted policymakers.
  • In the result and discussion section, some associated literature must be added to compare and contrast the key findings with the existing studies. Furthermore, Study limitations should be included in final conclusion part.
  • The conclusion should be based on your results and discussion. So, do consider it and improve it based on the logic of your results.
  • The conclusion does not properly describe as it was needful, hence please provide expansion in your conclusion-based estimations and provide some recommendations and policy implications more in detail.
  • The acronyms should be defined at first appearance in the manuscript and then must be consistently used throughout the manuscript. Furthermore, the manuscript must be checked for typo errors and spelling checks.

Reviewer 3 Report

Dear authors,

Your manuscript is interesting but I need you to answer some questions:

INTRODUCTION

  • The Introduction is very long. There should be no "Introduction" and "review" sections. The two sections must be joined into one.

MATERIALS AND METHODS

Study Phase:

  • The authors must specify the research design.
  • The authors provide redundant information.

DISCUSSION

  • The authors have not included study limitations.

CONCLUSIONS

  • The authors should simplify the conclusions. They are very long.

REFERENCES

  • Some references are incomplete or have errors. The authors should review this section.

Reviewer 4 Report

General comments:

This seems to be a well conducted study and its findings are potentially important in the control of the COVID-19 and other possible future disease outbreaks. However, I believe the presentation of the work could be improved considerably, and some important details of the method are missing (or are difficult to understand) in the current version of the manuscript.

Specific comments:

Abstract

It is not clear what “which was not controlled by human intervention” refers to.

The statement “Built environment is an important factor affecting the spread of the epidemic” is quite vague. Which aspects of the built environment?

The statement “However, different activity patterns in different age groups have different relationships with the built environment” could be rephrased for clarity. What sort of relationships?

Add "Covid-19" to “spatial distribution of patients”.

Should the fourth sentence read “regression analysis of the spatial distribution of patients and a classification of human behavior patterns”?

The abstract would be improved with further details of the results.

What “significant built environment factors” (last sentence)? This should be expanded/clarified.

Introduction

Paragraph 3: It would be useful to explain more about these “built environment factors” (what factors?) and how they have been studied in previous work. Built environment could mean many different things and at different spatial scales. (They are mentioned later in the Review section: “social and economic factors, land use factors, urban facilities factors, and urban open space factors”.) Specific examples of each type would also be helpful.

I think you need a bit more detail to explain why you have divided “the population into young and middle-aged people and the elderly according to their age and activity characteristics”. Why exactly do you expect differences between the age groups?

“The study period is the natural transmission stage before the artificial control”. Be more specific here.

Review

I think the following sentences are not entirely correct and too limited: “Public health focuses on the origin of the virus and the mechanism of infection [2-4], and there are still challenges for the development of specific vaccines [5]. Sociology focuses on people's anxiety, depression and social inequality in the epidemic period [6-7].” For example, public health is concerned also with methods for controlling the virus, ensuring adequate health service provision, etc.

This sentence needs referencing: “Research on the current situation of urban geography reveals the relationship be-tween health events and urban built environment and geographical space, and points out issues that have long been neglected in urban security issues in the planning field.”

The final sentence is very long and should be split into two sentences.

Materials and Methods

The first sentence is confusing: “Wuhan is a typical Asian mega-city with high density, and a relatively dense population of 11.21 million”. Is it high density or relatively dense? Does the first “density” refer to the built environment?

I found section 3.3 confusing. What is meant by “necessary and unnecessary activities”? How did you classify people by age based on the Weibo data? Do you have that information? One sentence says “patients are divided into young and middle-aged people and the elderly people through their behavior pattern” but I don’t not understand this. There is very little information about the specific built environment factors (which would be helpful at this point in the paper, even though they are described later) and the source(s) of this data. Important information seems to be missing, in particular the spatial resolution of the data. How precisely do you know the locations of the Weibo data? POI should be defined and explained.

OLS should be defined and explained in the Methods section. I assume it is ordinary least squares but this needs to be clear.

Results

I think it would be helpful to explain what each result means in more straightforward language. For example, “the kernel density of young and middle aged people seeking for help from Weibo showed that the kernel density of bus stations, the kernel density of middle schools and the volume ratio were positively correlated with the kernel density distribution” - does this mean that density of bus stations and middle schools and volume ratio were positively associated with help-seeking behaviour?

Do you need to say “kernel density” every time throughout the Results (and Discussion)? Could you instead state in the Methods that each factor is represented by its kernel density and then just refer to the factors?

The Table headings and Figure labels are unclear. In particular, they should state which age group they present.

Discussion

Throughout the Discussion explanations are suggested for the observed associations and differences between the two age groups. I think these should be stated less confidently because they seem to be hypothesised rather than demonstrated by the data (or from other published evidence). For example, “The kernel density of community-level shopping facilities is significant in the elderly population because the elderly cannot use the network skillfully and depend on the physical shopping facilities”. Although likely to be true, this is speculative and should be stated as such.

Section 5.5 should be more clearly linked to the research by referring directly to specific findings.

The Discussion needs a section on the limitations of the work (some limitations are very briefly mention at the end of the paper but I do not think this is sufficient). The implications of the key limitations/uncertainties should be discussed. How likely are they to influence your findings and conclusions?